# PROCEEDINGS A

complexity

fractal network structure, time constraint, trust, network fragmentation, morbidity, mortality

**Author for correspondence:**
R. I. M. Dunbar
e-mail: robin.dunbar@psy.ox.ac.uk

# Structure and function in human and primate social networks: implications for diffusion, network stability and health

## R. I. M. Dunbar

Department of Experimental Psychology, University of Oxford, New Radcliffe Building, Radcliffe Observatory Quarter, Oxford OX1 6GG, UK

RIMD, 0000-0002-9982-9702

The human social world is orders of magnitude smaller than our highly urbanized world might lead us to suppose. In addition, human social networks have a very distinct fractal structure similar to that observed in other primates. In part, this reflects a cognitive constraint, and in part a time constraint, on the capacity for interaction. Structured networks of this kind have a significant effect on the rates of transmission of both disease and information. Because the cognitive mechanism underpinning network structure is based on trust, internal and external threats that undermine trust or constrain interaction inevitably result in the fragmentation and restructuring of networks. In contexts where network sizes are smaller, this is likely to have significant impacts on psychological and physical health risks.

## 1. Introduction

The processes whereby contagious diseases or information propagate through communities are directly affected by the way these communities are structured. This has been shown to be the case in primates [1–3] and has been well studied in humans in the form of epidemiological [4] and information diffusion (opinion dynamics or voter) models [5]. The Ising phase state model (originally developed to describe the magnetic dipole moments of atomic spin in ferromagnetism) has

been the workhorse of most of these models and of many of the models currently used to calculate the value of the *R*-number (or reproduction rate) used to drive current COVID-19 management strategies.

Most early models were mean field models that assumed panmixia. However, human social networks are highly structured and small world: most people interact with only a very small number of individuals whose identities remain relatively stable over time. When it became apparent that the structure of networks could dramatically affect the flow of information (or infections) through networks [6,7], structure began to be incorporated into epidemiological models [8–12]. Many of the best current models are 'compartmental models' which represent structure by the fact that a community consists of households or other small population units [11,12]. In effect, these use spatial structure as a proxy for social structure, which has the advantage of ensuring that the models compute easily. In reality, of course, it is people's interactions with each other that give rise to the spatial structure represented by households. While it is true that most (but not all) individuals see and interact with household or family members more often than with anyone else, in fact this dichotomizes what is in reality a continuum of interaction that flows out in ripples from each individual. These ripples create social layers of gradually declining amplitude that spread through the local community well beyond the household.

My aim in this paper is to examine the social and psychological processes that underpin natural human sociality in order to better understand how these affect both network structure and the way information or diseases propagate through them. Like all monkeys and apes, humans live in stable social groups characterized by small, highly structured networks. Individuals do not interact with, let alone meet, everyone else in their social group on a regular basis: a very high proportion of their interactions are confined to a very small subset of individuals. These relationships are sometimes described as having a 'bonded' quality: regular social partners appear to be fixated on each other [13,14]. The mechanisms that underpin these relationships have important consequences for the dynamics of these networks.

I will first briefly review evidence on the size and structure of the human social world. I will then explain how the cognitive and behavioural mechanisms that underpin friendships in all primates give rise to the particular form that human networks have. Finally, I explore some of the consequences of this for information and disease propagation in networks, and how networks respond to external threats.

## 2. Dunbar's number: a small social world

Humans have lived in settlements only for the past 8000 years or so, with mega-cities and nation states being at all common only within the last few hundred years. Prior to that, our entire evolutionary history was dominated by very small-scale societies of the kind still found in contemporary hunter–gatherers. Our personal social worlds still reflect that long evolutionary history, even when they are embedded in connurbations numbering tens of millions of people. Table 1 summarizes the sizes of egocentric personal social networks estimated in a wide variety of contexts ranging from Xmas card distribution lists (identifying all household members) to the number of friends on Facebook, with sample sizes varying between 43 and a million individuals. The mean network size varies across the range 78–250, with an overall mean of approximately 154.

Table 1 also lists a number of studies that have estimated community size in a variety of pre-industrial societies as well as some contemporary contexts where it is possible to define a personalized community within which most people know each other on a personal level. These include the size of hunter–gatherer communities, historical European villages from the eleventh to the eighteenth centuries, self-contained historical communes, academic subdisciplines (defined as all those who pay attention to each other's publications) and Internet communities. The average community sizes range between 107 and 200, many with very large sample sizes, with an overall mean of approximately 158.

**Table 1.** Social network size in humans.

| grouping | sample | size | source |
|---|---|---|---|
| egocentric network | | | |
| small world network | 2 expts | 134 | [15] |
| Christmas card distribution list | 43 | 153.5 | [16] |
| women's egocentric networks | 251 | 71.8 | [17] |
| egocentric network | 339 | 174.9 | [18] |
| cellphone calling network (Europe) | 26 680 | 134.3 | [19] |
| cellphone calling network (China) | 15 209 | 141.4 | [20] |
| US wedding guest list | 10 year mean | 143.7 | [21] |
| E-mail networks | 35 600 | 250 | [22] |
| Facebook friends (mode) | 1 000 000 | 150–250 | [23] |
| Facebook friends (UK, 2 samples) | 3375 | 169 | [24] |
| co-author networks (sciences) | 285 577 | 116.8 | [25] |
| average | | 153.6 ± 46.0 s.d. | |
| community size | | | |
| Domesday Book (1087 AD) (mean village size) | | 150 | [26] |
| C18th English villages (mean size) | | 160 | [27] |
| Italian alpine communities (1250–1800 AD) | | 176 | [28] |
| tribal societies (community size) | 9 | 148 | [29] |
| hunter–gatherer societies (clan size) | 339 | 165 | [30] |
| E. Tennessee rural community | 1 | 197 | [31] |
| Hutterite farming communities | 51 | 107 | [32] |
| 'Nebraska' Amish parishes | 8 | 113 | [33] |
| Church congregations (ideal size) | | 200 | [34] |
| company (mean, WW2 armies) | 10 | 180 | [35] |
| academic research specialities | 13 | 100–200 | [36] |
| Twitter networks | 60 790 | 100–200 | [37] |
| average | | 158.0 ± 28.7 s.d. | |

The value of approximately 150 as a natural grouping size for humans was, in fact, originally predicted from an equation relating social group size to relative neocortex size in primates before this empirical evidence became available [38]. This prediction had a 95% confidence interval of 100–250, very close to the observed variance in the data. In primates as a whole (but not other birds and mammals), social group size is a function of neocortex volume, and especially the more frontal neocortex regions (the Social Brain Hypothesis [39]). In the last decade, neuroimaging studies of both humans [40–50] and monkeys [51,52] indicate that the relationship between personal social networks (indexed in many different ways) and brain size also applies within species at the level of the individual as well as between species.

The social brain relationship arises because primates are unusual in that they live in relatively large, stable, bonded social groups [53]. In contrast with the more casual groups (i.e. herds, flocks) of most mammals and birds, the problem with bonded groups is that they are, in effect, a version of the coupled oscillator problem. If animals' foraging and resting schedules get out of synchrony,

some individuals will drift away when others go to rest, resulting in the fragmentation of the group [54,55]. Individuals have to be willing to accept short-term costs (mainly in relation to the scheduling of foraging) in order to gain greater long-term benefits (protection from predators by staying together). Maintaining spatial coherence through time is cognitively difficult. It depends on two key psychological competences that appear to be unique to the anthropoid primates: the ability to inhibit prepotent actions (a prepotent response is the tendency to take a small immediate reward in preference to waiting for a larger future reward) and the capacity to mentalize. Inhibition depends on the volume of the brain's frontal pole [56], while mentalizing depends on a dedicated neural circuit known as the theory of mind network (also known as the default mode neural network) that integrates processing units in the brain's prefrontal, parietal and temporal lobes [57], supplemented by connections with the limbic system [41,42]. The frontal pole is unique to the anthropoid primates [56]; the default mode network that underpins mentalizing is also common to both humans and monkeys [58].

Maintaining group cohesion is not simply a memory problem (though it is commonly misunderstood as such). Rather, it is one of predicting others' future behaviour under different conditions (e.g. knowing how others will respond to one's own actions) and being able to gauge the future reliability (trustworthiness) of other individuals [38,59]. This is much more costly in terms of both neural activity and neural recruitment than simple factual recall [60]. In humans, the number of friends is directly correlated with mentalizing skills [44,61], and mentalizing skills are, in turn, correlated with the volumetric size of the brain's default mode neural network [62,63]. The latter relationship has recently been shown to extend to primates as a whole [64].

## 3. Social networks and Dunbar graphs

Social networks have generally been viewed from two different perspectives. Network analysts with a statistical physics background have tended to view them top-down as macroscopic phenomena (i.e. from above looking down on the spatial distribution of a population of nodes), whereas sociologists have tended to view them from below as egocentric networks (the individual's experience of that population). On the whole, the first group have tended to focus on large-scale patterns in very large networks, often with an emphasis on triadic closure (Heider's structural balance theory [65]) as the glue that gives structure to a network; the second have focused on the micro-structure of individual's personal social networks, often focusing on the inner core of intimate friendships immediately beyond the simple triad.

An important finding from the second approach has been that networks actually consist of a series of layers that correspond to relationships of different quality [16,59,66]. Seen from an egocentric point of view, the frequency with which an individual contacts the members of their network does not follow a conventional power law distribution but, on closer inspection, contains a series of plateaux. Cluster analyses of very large datasets invariably reveal that these personal networks contain four layers within them (figure 1). This gives the network a layered structure, where individual alters in a given layer are contacted with more or less similar frequency and there is a sharp drop-off in contact frequencies to the next layer. It turns out that, while there is some individual variation, these layers have quite characteristic sizes. Moreover, when counted cumulatively, they have a very distinct scaling ratio: each layer is approximately three times the size of the layer immediately inside it (figure 2).

This layered structure in figure 2 (referred to as a Dunbar graph [67]) has been identified, with virtually the same numerical layer values, in surveys of network size, the calling patterns in national cellphone databases, science co-author networks and the frequencies of reciprocated postings in both Facebook and Twitter (table 2). Each layer seems to correspond to a very specific frequency of interaction (figure 3), and these frequencies are remarkably consistent across media [66], suggesting both that they are hardwired and that communication media are substitutable. One way this structure might arise would be if the basal layer of five people represented, for example, a family or household, such that the next layer of 15 consists of three families with an especially close relationship, and the 50-layer beyond that consisted of three of these trios. This

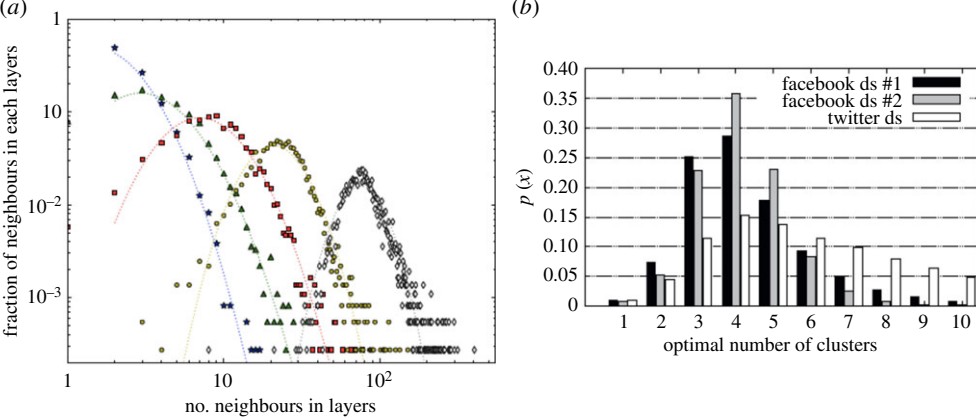

**Figure 1.** (*a*) Layers within a large European cellphone dataset, based on frequency of calling. Layers identified by Jenks clustering algorithm. Reproduced from [19]. (*b*) Optimal number of clusters identified by *k*-means clustering algorithm in three online datasets. Reproduced from [68].

pattern, however, is likely to reflect small-scale traditional communities; in more mobile, post-industrial societies, figure 2 can arise simply a consequence of the patterns of interaction between individuals and need have no family-based (or spatial) underpinning to it at all. It is notable, nonetheless, that the same patterns emerge in either case, suggesting that there is an underlying universal constraint on how networks are constructed.

This same pattern, with the same layer sizes, has also been identified in a number of top-down analyses of the social (or spatial) organization of human populations (table 2), including hunter–gatherers, the size distribution of Irish Bronze Age stone circles (as estimates of local population distribution), the sizes of residential trailer parks, the structure of modern armies, the size of communities of practice in the business world and even the patterns of alliance formation in massive online multiplayer games (MOMs). This pattern, with the same scaling ratio, has also been noted in the political organization of a large sample of historical city states [69]. In fact, this layered structure with a consistent scaling ratio was first noted in an analysis of hunter–gatherer societies in the early 1980s by Johnson [70], who suggested that it was a 'span on control' management solution in response to internal stresses created as grouping size increases.

More surprisingly, perhaps, these same numbers reappear in both the distribution of primate social group sizes [71] and in the layered structure of groups for those mammals that live in multilevel social systems (mainly baboons, chimpanzees, elephants and dolphins) [72,73] (table 2). Animal societies with stable groups do not extend beyond the 50-layer, but all the internal layers are present. The fact that these numbers are so consistent across so wide a range of species perhaps suggests that they may be the outcome of natural scaling effects due to the structural stresses that social groups incur as they grow in size in response to ecological demands. As a result, social evolution in primates [74] occurs as a result of a stepwise increase in group size [71] achieved by bolting together several basal subgroups to create successive layers rather than through a continuous adjustment of group sizes as in most birds and mammals. Primate species achieve larger groups by delaying group fission that would normally act as a nonlinear oscillator to keep group size within a defined range around the local mean [81–83]. The process thus seems to behave more like a series of phase transitions triggered by a natural fractionation process.

Although, in humans, there is remarkably little variation in both overall network size and layer sizes across samples, irrespective of sample size, sample origin and cluster detection algorithm, nonetheless within populations, there is considerable variation between individuals (figure 1*a*). Some of this variation is due to sex (women tend to have larger inner layers than men [59,60,84],

**Table 2.** Mean layer sizes from different datasets.

| | sample size size | layer | | | | | | | clustering algorithm | source |
|---|---|---|---|---|---|---|---|---|---|---|
| | | 1 | 2 | 3 | 4 | 5 | 6 | 7 | | |
| personal networks | | | | | | | | | | |
| close network members | 101 | | 4.7 | 11.9 | | | | | survey[a] | [66] |
| Xmas card network | 43 | | 7 | 21 | 35 | 153.5 | | | natural breaks | [16] |
| complete social network | 339 | | 6.1 | 22.6 | 174.9 | | | | survey[a] | [18] |
| cellphone network (Europe): | | | | | | | | | | |
| frequency of calls | 26 680 | 2.9 | 7.4 | 17.7 | 43.0 | 134.3 | | | Jenks | [19] |
| duration of calls[c] | 26 680 | | 3.9 | 10.1 | 27.2 | 129.3 | | | Jenks | [19] |
| cellphone network (China) | 15 209 | 2.1 | 7.3 | 20.4 | 54.2 | 141.4 | | | k-means | [20] |
| Facebook network no. 1 | 130 338 | 1.7 | 5.3 | 14.9 | 40.9 | | | | k-means | [68] |
| Facebook network no. 2 | 60 290 | 1.5 | 4.3 | 10.7 | 27.0 | | | | k-means | [68] |
| Facebook network no. 3 | 3375 | | 4.1 | 13.6 | 169.0 | | | | surveys[b] | [24] |
| co-author networks | 285 577 | 2.0 | 6.3 | 15.8 | 37.9 | 116.8 | | | k-means | [23] |
| average | | 2.0 | 5.8 | 15.8 | 34.6 | 148.3 | | | mean scaling ratio = 3.0 | |
| community structure | | | | | | | | | | |
| hunter–gatherer societies no. 1 | 61 tribes | | | | 37.7 | 147.8 | | 1154.7 | spectral analysis | [75] |
| hunter–gatherer societies no. 2 | 339 tribes | | 4.5 | 15.6 | 53.5 | 165.7 | (837.1) | | Horton order | [30] |
| communities of practice | 130 CoPs | | 4.0 | 11.0 | 30.2 | 112.2 | 389.0 | 1737.8 | k-means | [76] |
| Shenzhen 100 investor network | 381 345 | | | 10.0 | 68.8 | 142.9 | | | k-means | [20] |
| residential trailer parks sizes[d] | 53 sites | | | 16.2 | 56.4 | 139.6 | 350.0 | 677.2 | Jenks | [77] |

(*Continued.*)

**Table 2.** (*Continued.*)

| | sample size size | layer | | | | | | | clustering algorithm | source |
|---|---|---|---|---|---|---|---|---|---|---|
| | | 1 | 2 | 3 | 4 | 5 | 6 | 7 | | |
| MoM (*Pardus*) | 380 000 players | 1.6 | 5.1 | 11.5 | 24.7 | 294 | — | 1832 | Horton order | [78] |
| Twitter communities | 60 790 | 1.6 | 4.5 | 11.2 | 28.3 | 88.3 | | | *k*-means | [68] |
| Irish stone circles^e | 140 | 5 | 14 | 64 | 145 | | | | natural breaks | [79] |
| modern armies^f | | | (4–5) | 5–14 | 15–45 | 80–150 | 300–800 | ~1500 | | [80] |
| average (excluding military) | | 1.6 | 4.6 | 12.8 | 45.5 | 154.4 | 369.5 | 1350.4 | mean scaling ratio = 3.11 | |
| mammals | | | | | | | | | | |
| multilevel mammals (layer size)^g | | 3.1 | 6.9 | 19.8 | 47.3 | | scaling ratio = 2.49 | | Horton order | [72] |
| primate species (group sizes)^h | 2.3 | 6.2 | 15.6 | 31.3 | 53.1 | | scaling ratio = 2.23 | | Jenks | [71] |

^a Surveys based on enumerating size of layers defined by specific temporal or emotional closeness criteria identified with these layers.
^b Average of two national, stratified UK surveys ($N_1 = 2000$ and $N_2 = 1375$).
^c Excluded from calculation of average values.
^d Actual occupancy at the time of survey; a larger sample of 1216 sites estimating maximum occupancy from average occupancy and total pitches yields similar values.
^e Local community size estimated from the internal area of the circle, allowing a constant social distance between neighbours.
^f Range of unit sizes in modern national armies. Layer 1 corresponds to special forces units; subsequent layers correspond to (2) section, (3) platoon, (4) company, (5) battalion and (6) regiment; and continue beyond this as brigade (approx. 5000), division (approx. 15 000) and corps (approx. 50 000).
^g Mean of species mean layer values. Four species were included in the sample: elephant (*Loxodonta africanus*: $N = 312$ individuals), orca (*Orcinus orca*: $N = 216$), gelada (*Theropithecus gelada*: $N = 1036$) and hamadryas baboon (*Papio hamadryas*: $N = 742$).
^h Complete sample of $N = 215$ species for which the mean species group size is known.

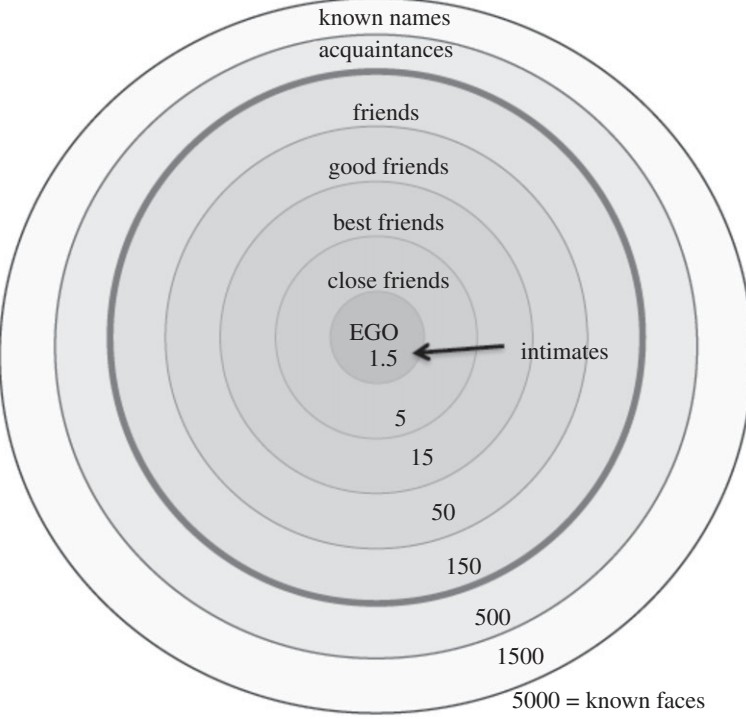

**Figure 2.** Structure of human egocentric social networks. The number of people included in each circle increases, but the frequency of contact and emotional closeness declines, with each layer. Redrawn from [19]. The outermost layer (5000) was identified by experimental studies of face recognition (the number of faces that can be recognized as known by sight) [80]. (Online version in colour.)

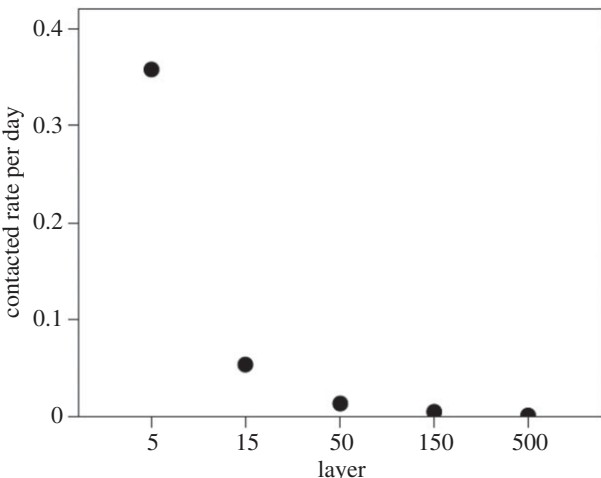

**Figure 3.** Frequency per day with which people contacted individual members of each layer in their social network. Sample: complete social networks of 251 UK and Belgian women. Reproduced from [17].

but smaller outer layers [84]), some due to age (network and layer sizes are an inverted-J-shaped function of age, peaking in the 20s–30s [16,24,86]) and personality (extroverts have larger networks at all layer levels than introverts [85–88]).

In addition, all human social networks are divided into two components, family and friends. Although in small-scale societies, virtually everyone in your network is a member of your extended family, in the post-industrial world with our lower birth rates, the typical network is split roughly 50 : 50 between family and friends [59,66,89]. However, it still seems that preference is given to family: those who come from large extended families have fewer friends [17,66]. In effect, family and friends form two largely separate subnetworks that are interleaved through the layers of the network, with a roughly even split in the two innermost 5- and 15-layers, friends predominating in the middle 50-layer and family predominating in the outer 150-layer [59]. The latter seems to reflect the fact that friends are more costly to maintain in terms of time investment than family members [17], and hence survive less well in the outermost layer (see below).

Conventional top-down networks tend to focus on the degree of individual Ego's, usually with some kind of cut-off to define how many primary contacts an individual has. Irrespective of where the cut-off is taken to be, these relationships tend to be viewed as one-of-a-kind. Dunbar graphs, by contrast, recognize that individuals have relationships of different quality with many individuals, which might be viewed as primary, secondary, tertiary, etc. relationships. The first will usually be a subset of the second.

## 4. Time, trust and the bonds that bind

In this section, I provide a brief explanation of how the primate bonding process works. The main purpose is to stress that it is both complex and time-consuming. This will help explain some of the patterns we will meet in the following two sections where I discuss network dynamics and their consequences.

Primate social groups are implicit social contracts designed to ensure protection from predators and, secondarily, rival conspecifics through group augmentation selection effects [90]. Group-living is often mistaken for a cooperation problem, but it is in fact a coordination problem. Cooperation problems invariably involve a public goods dilemma (cooperators pay an up-front cost), whereas a coordination problem does not (you are either in the group or not, and everyone pays the same simultaneous cost) [91]. The problem animals have to solve is how to maintain group stability (i.e. coordination) in the face of the stresses that derive from living in close proximity [81,83,92] which would otherwise cause the members to disperse (as happens in herd- and flock-forming species [54,55]. The primate solution to this problem is bonded relationships, since this ensures that individuals will maintain behavioural synchrony and stay together as a group. In primates, relationships are built up by social grooming [93]. Grooming is an exceptionally time-costly activity, with some species devoting as much as one-fifth of their entire day to this seemingly trivial activity [93]. Grooming creates a sense of reciprocity, obligation and trust (operationally, a form of Bayesian belief in the honesty and reliability of an alter's future behaviour [94]).

The layered structure of human social networks is a consequence of how we choose to distribute our social time around the members (figures 2 and 3). In both monkeys [92] and humans [59,94], the quality of a relationship (indexed by its likelihood of producing prosocial support when it is needed) depends directly on the time invested in it. However, the time available for social interaction is severely limited [17,85,95], and this forces individuals to make trade-offs between the benefits offered by relationships of different strength and the costs of maintaining those relationships.

The process involved is a dual process mechanism that involves two separate mechanisms acting in parallel. One is a psychopharmacological mechanism acting at the emotional (raw feels) level that is mediated by the way social grooming triggers the brain's endorphin system (part of the pain management system); the other is the cognitive element that forms the core of the social brain itself [96].

Social grooming is often assumed to have a purely hygienic function. While it certainly has this effect, in primates it has been coopted to form a far more important function in social bonding. The action of leafing through the fur during grooming triggers the endorphin system in the brain [97,98] via a specialized neural system, the afferent C-tactile (or CT) fibres [99]. These are highly specialized nerves that respond *only* to light, slow stroking at approximately 2.5 cm s$^{-1}$ (the speed of hand movements in grooming), and nothing else [100]. Endorphin activation and uptake in the brain creates an opioid-like sense of relaxation, contentment and warmth [101,102] that seems to provide a psychopharmacological platform for bondedness [101,103–105] off which the second process, a cognitive sense of trust and obligation, is built. Endorphins have a relatively short half-life (around 2.5 h), and so the system needs constant activation via grooming to maintain the requisite bonding levels, thereby making the system very time-costly.

Physical touch in the form of stroking and casual touch continues to form an important part of human social interaction and yields exactly the same endorphin effect [98] as it does in primate grooming. However, physical contact has an intimacy that limits it mainly to the inner layers of our networks [106,107]. Moreover, it is physically impossible to groom with more than one individual at a time with the same focused attention that seems to be important in its execution. This, combined with the constraints on time and the minimum time required to maintain a working relationship, ultimately places a limit on the number of relationships an animal can bond with using only this mechanism. In primates, this limits group size to about 50 individuals (the upper limit on average species group size in primates [68]). Groups larger than this are prone to fragmentation and ultimately to fission [108].

In order to be able to increase group size, humans discovered how to exploit other behaviours that also trigger the endorphin system in a way that is, in effect, a form of grooming-at-a-distance. These include laughter [109], singing [110], dancing [111], emotional storytelling [112], and communal eating [113,114] and drinking (of alcohol) [115], all of which trigger the endorphin system and do so incrementally when done in synchrony [116,117]. Because they do not involve direct physical contact, more individuals can be 'groomed' simultaneously, thereby allowing a form of time-sharing that makes it possible to reach more individuals and so increase group size.

The second component of this system is a cognitive mechanism. It centres around the knowledge of other individuals that can be built up by being in close contact. Evolutionary studies of cooperation tend to view relationships as a form of debt-logging. Such knowledge would not necessarily require frequent interaction since that can be done by third-party observation. Rather, interacting closely with others allows an individual to get to know them well enough to predict their behaviour—to know that they really will come to your aid when you really need them, not because they owe you a favour but because they have a sense of obligation and commitment to you. In effect, it creates a sense of trust that acts as a rapid, intuitive (albeit imperfect) cue of reliability.

In humans, this has been augmented by a capacity to build a more detailed 'picture' of another individual through conversation in a way that short circuits the need to invest excessive amounts of time in getting to know them. In other words, we can form a near-instantaneous impression of a stranger and use that as the basis for decisions about whether or not to engage with them. We do this by evaluating an individual's status on a set of largely exogenous cultural dimensions known as the Seven Pillars of Friendship [96]. The seven pillars are: language (or, better still, dialect), place of origin, educational trajectory, hobbies and interests, worldview (religious/moral/political views), musical tastes and sense of humour. The more of these we share in common with someone, the stronger the relationship between us will be and the more altruistic we will be to each other [118]. This 'birds of a feather flock together' phenomenon is termed *homophily* [119].

The Seven Pillars are cues of membership of the same community. In small-scale societies, they would identify an extended kinship group, where kin selection (the 'kinship premium' [120]) provides an additional guarantee of trustworthiness [94]. In effect, they function as a cultural totem pole at the centre of the metaphorical village green on which the community can hang its hats—an emotional consensus of who we are as a community and how we came to be that

way, a way of building up mutual trust. In the contemporary context, it still identifies your (now reduced) kin group, but it also seems to identify the small community where you spent your formative years—the period when you acquired your sense of who you are and what community you belong to. This is the community whose mores and behaviour you understand in a very intuitive way, and it is this understanding that determines your sense of how well you can trust its members.

Homophily in friendships has also been documented in respect of a number of endogenous traits, including gender [17,24,66,121], ethnicity [122] and personality [85,123]. Gender has a particularly strong effect: approximately 70% of men's social networks consist of men, and approximately 70% of women's networks consist of women, with most of the opposite sex alters in both cases being extended family members. One reason for this is that the two sexes' style of social interaction is very different. Women's relationships are more dyadic, serviced mainly by conversation and involve significantly more physical touch, whereas men's relationships are more group-based, typically involve some form of activity (sports, hobbies, social drinking) rather than conversation, and make much less use of haptic contact [106,124,125]. Men's friendships are also typically less intense, more casual and more substitutable than women's: women will often try to keep core friendships going long after one of them has moved away (e.g. through e-mail or Facebook), whereas men simply tend to find someone else to replace the absent individual(s) in a rather out-of-sight-out-of-mind fashion.

Homophily enables interactions (and hence relationships) to 'flow', both because those involved 'understand' each other better and because they share strategic interests. Between them, these emotional and cognitive components create the intensity of bonds that act as the glue to bind individuals together. Human, and primate, friendships are often described in terms of two core dimensions: *being close* (desire for spatial proximity) and *feeling close* (emotional proximity) [126]. Between them, these ensure that bonded individuals stay together so that they are on hand when support is needed, thereby ensuring group cohesion through a drag effect on other dyads created by the ties in the network. Since bonding, and the time investment this is based on, is in principle a continuum, this will not, of itself, give rise to the layered structure of a Dunbar graph. To produce this, two more key components are needed: a constraint on time and the fact that friendships provide different kinds of benefits. The next section explains how this comes about.

## 5. How time structures networks

Two approaches have been used to understand why social networks might have the layered structures they do. One has been to use agent-based models to reverse engineer the conditions under which the intersection between the time costs of relationships and the benefits that these provide give rise to structured networks. The other has been to solve the problem analytically as an optimal investment decision problem. In many ways, these represent top-down and bottom-up approaches, with the first focusing on the macro-level exogenous conditions that produce layered networks, and the second focusing on the micro-decisions that individuals have to make within these environments when deciding whom to interact with.

Building on the time budget models of Dunbar *et al*. [95], Sutcliffe *et al*. [127] conceived the problem as the outcome of trying to satisfy two competing goals (foraging and socializing, where socializing provides a key non-foraging ecological benefit such as protection against internal or external threats) in a time-constrained environment. The aim was to identify which combination of strategies and cost/benefit regimes reproduced the exact layer structure of human social networks (for these purposes, taken as the 5, 15 and 150 layers) when agents try to maximize fitness (with fitness being the additive sum of five proximate components that reward social strategies differentially). In the model, 300 agents armed with different strategic preferences for investing in close versus distant ties interacted with each other to form alliances that provided access to different resources. In each run, the population initially consisted of equal numbers of agents pursuing each of three investment preference strategies. Following the conventional design for models of this kind, at the end of each round (generation) the 20% of agents with the

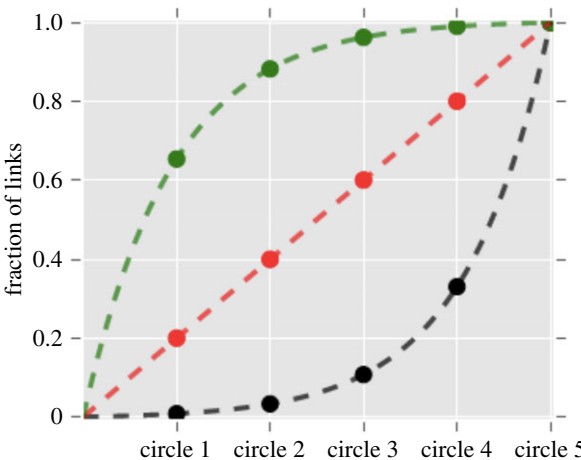

**Figure 4.** Cumulative proportional composition across layers of egocentric networks predicted by a one-dimensional Bayesian decision model. When the Lagrange multiplier $\mu < 0$, the network structure reverses from the conventional structure (upper curve) to a reversed structure (lower curve). Reproduced from [128]. (Online version in colour.)

lowest fitness were removed (died) and were replaced by offspring produced by the top 20% (duplicating the parent's strategy), thereby allowing the size of the population to remain constant but its composition to evolve. The model was allowed to run until the distribution of strategy types reached an equilibrium (typically 2000 cycles). The outcomes from greater than 3000 runs (in which weightings on the fitness functions were systematically varied) were sorted into clusters using $k$-mean cluster analysis with quantitative fit to the Dunbar numbers as the criterion.

Numerically, strategies that favour having just a few strong ties dominate the fitness landscape, yielding the kinds of small monogamous or harem-based societies that are widely common in mammals. The second most common strategy was the one where agents have no preferences for forming close ties of any kind and instead form large, anonymous herds with no internal structure similar to those characteristic of herd-forming mammals. By contrast, multilevel social structures of the kind found in many primates, and especially those with the specific layer sizes of human social networks, were extremely rare (accounting for less than 1% of runs). They occurred only under a very limited set of circumstances, namely when a capacity for high investment in social time (i.e. sufficient spare time over that needed for foraging), preferential social interaction strategies, high mortality risk and steep reproductive differentials between individuals coincided. These are all traits characteristic of primates.

The alternative approach considers the investment decisions themselves. Tamarit *et al.* [128] developed a one-parameter Bayesian statistical urn model in which individuals choose to invest their limited social capital in relationships that provide different benefits (identified here as network layers). The model seeks to optimize the distribution of effort across relationship types

$$P(\ell|\mathcal{S}, \mathcal{L}, N) = B(L, \mathcal{L}/N, N) \binom{L}{\ell} \frac{e^{-\mu \sum_k s_k \ell_k}}{(\sum_k e^{-\mu s_k})^L},$$

where $s_k$ is the cost of a relationship in layer $k \in 1, 2 \ldots r$, $\ell_k$ is the number of alters assigned to layer $k$ (with $L = \Sigma l_k$), $S$ is the total amount of resource (time) available, $N$ is the total population size and $\mu$ is the Lagrange multiplier associated with the constraint imposed by total resources. When the cost (i.e. the time investment that has to be made in a relationship) is a monotonic negative function of layer (as in figure 3), this yields layered structures of exactly the kind found in human social networks—few close ties and many weak ones (the lower curve in figure 4).

However, it turned out that the structure of the network inverts (more close ties and few distant ties) when the Lagrange multiplier falls below $\mu = 0$ (upper curve in figure 4). One context in

which this might happen is when the available community of alters to choose from is small, and so there is spare resource capacity per available alter. A comparison of migrant versus host communities in Spain revealed that there is indeed a phase transition between conventionally structured networks (layers with a concave structure, i.e. a few close friends, more intermediate friends and many casual friends) to inverted networks (convex structures, in which Ego has more close friends, some intermediate friends and only a few casual friends) at $\mu \approx 0$ [128]. This had not been noted previously because in the normal population, most individuals fall in the $\mu > 0$ region; the very small number falling in the $\mu < 0$ region had simply been viewed as statistical error. However, most migrants, who typically have fewer social opportunities, fall in the $\mu < 0$ region and so have inverted (convex) networks as a result, with only a small number having standard (i.e. concave) networks. This seems to suggest that, when surplus time is available, people prefer to invest more heavily in their closest ties rather than distributing their effort more evenly because the emotional and practical support offered by these inner layer ties is more valuable to them. This may explain the structure of primate groups where the inner core of strong ties typically represents a larger proportion of group size in species that live in smaller groups.

The fact that the scaling ratio is consistently approximately 3 in human (and primate) social networks raises the possibility that Heider's structural balance triads might explain the layered structuring. This possibility was considered by Klimek *et al.* [129] who studied an Ising-type coevolutionary voter model in a context where there are several social functions (say, friendship and business alliances) such that there are separate linked networks where most individuals occupy positions in each network. They showed that when the networks (i.e. functions) vary in rewiring probability (slow versus fast turnover in relationships), the single large network will eventually undergo a phase transition known as *shattered fragmentation* in which the community fractures into a large number of small subnetworks (or cliques). This happens only when one of the rewiring frequencies reaches a critically high level.

When Klimek *et al.* examined data from the *Pardus* online MOM game world, they found that the slow rewiring network (friendship) produced a weakly multimodal right-skewed, fat-tailed distribution with modal group sizes at 1–2 and approximately 50 players with a few very large super-communities centred around 160 or 1200 members. By contrast, the two fast rewiring networks (in this context, trading and communication functions) both underwent fragmentation into a large number of smaller subnetworks, with a single peak in group size at approximately 50 in both cases, just as the model predicts. When the three networks were projected onto a single multidimensional mapping, very distinct peaks emerged in the distribution of group sizes in both model and data at approximately 40, 150 and 1200, much as we find in table 2. This seems to suggest that when triadic closure is a criterion for relationship stability and there is more than one criterion by which individuals can form ties, layering emerges naturally in networks through self-organizing fragmentation effects.

So far, we have considered hierarchically inclusive layer structures. In these, the whole population is contained in the lowest layer, and the higher layers are created by successively bolting together, layer by layer, the groupings that make up each lower layer rather in the way military units are structured [130]. Most social networks seem to work like this. However, layers can also arise when some individuals are allocated positions of status, so that the members of the community are distributed across different layers with most individuals in the base layer and a few individuals in one or more higher layers. Networks of this kind are characteristic of management structures and the kinds of social hierarchies found in feudal societies.

Layered structures of this kind seem to emerge rather easily when individuals differ in their willingness to compromise. Dávid-Barrett & Dunbar [131] used an agent-based model to investigate the processes of group coordination when a community has to converge on an agreed compass direction (a proxy for any communal action or opinion that has the advantage of allowing up to 360 different values to be held rather than just two as in more conventional Ising models), but one group member is so convinced they have the right answer that they refuse to compromise. If agents can assign weightings to each other on the basis of some preference criterion, however arbitrary, a layered structure emerges with an 'elite' subgroup that acts, in

effect, as a management clique. Multilevel structures of this kind have the advantage that they increase the speed with which decisions are adopted. Multilayer networks are optimal when the costs associated with maintaining relationships, combined with the costs of information flow, are high. In such cases, a social hierarchy can be adaptive: when the hierarchy is steep, information needs to traverse fewer relationships (shorter path lengths), either because the elite effectively act as bridges between lower level groups (distributed management) or because the elite imposes its decisions on the individuals in the lower strata (dictatorial management). Falling communication costs lead to a less steep hierarchy when socially useful information is evenly distributed, but to a steeper hierarchy when socially useful information is unevenly distributed.

## 6. Propagation in structured networks

In human social networks, the layers have very characteristic interaction frequencies with Ego (figure 3). Approximately 40% of all social effort (whether indexed as the frequency or duration of interaction) is directed to the five individuals in the closest layer, with another 20% devoted to the remaining 10 members of the second layer. Thus, 60% of social time is devoted to just 15 people. Comparable results have been reported from large-scale surveys in the UK [8] and in China [10]. This will inevitably affect the rate with which information, innovations or disease propagate through a network. However, network structure can speed up or slow down the rate of propagation, depending on the precise nature of the social processes involved.

In a very early (1995) analysis of this, we used Boyd & Richerson's [5] mean field Ising model of cultural transmission to study what happens when these kinds of models are applied to structured networks [130]. In the model, individuals acquire their information from *n* cultural parents, each of whom can differ in their cultural state. The model was run with a population of 10 000 agents mapped as nodes on a 100 * 100 lattice wrapped on a torus so as to prevent edge effects. Structure was imposed by allowing nodes to interact only with their eight closest neighbours on the lattice. On a regular lattice, these consist of two distinct sets: direct contacts (the four adjacent nodes on the main diagonals) and indirect contacts (the four corner nodes that can only be reached indirectly through the four adjacent nodes). In effect, these are equivalent to friends and friends-of-friends. At each generation, a node can change its cultural variant either by mutation or by imitation from one of its neighbouring nodes, with transition probabilities determined by a three-element vector specifying node-specific values of the Boyd–Richerson cultural inheritance bias functions (one reflecting the self-mutation rate, the other two the transmission, or copying, rates from the four 'friends' and the four 'friends-of-friends', with the proviso that all three sum to 1).

When the spatial constraint is dropped and everyone is allowed to interact with everyone else, the model replicates exactly the findings of the Boyd–Richerson [5] cultural inheritance model. The population evolves to full penetrance by a mutant cultural variant initially seeded at just one node (i.e. with a probability of occurrence of just 0.0001) in 75–150 generations. With spatial (or social) constraints in place, however, two important effects emerge. First, depending on the steepness of the inheritance bias functions, 44–60% of mutant seedings went extinct before achieving full penetrance, apparently because they often became trapped in eddies at particular locations and could not break out before going extinct. In those runs where the mutant achieved full penetrance (i.e. all nodes became mutants), the time to penetrance was 150–300 generations for the same set of transmission biases. In other words, the mutant trait took far longer to spread through the population.

Once again, the time taken to break out of local eddies was the main reason for the much slower penetrance. The difference between these runs and those where the mutant went extinct depended on the balance between the stochastic rates at which new 'infected' clusters were created and died out. If a local extinction occurred early in the system's evolution when few mutant clusters had become established, global extinction was more likely (the classic small population extinction risk phenomenon in conservation biology [132]). Changing population size or the number of cultural 'parents' had a quantitative effect, but did not change the basic pattern.

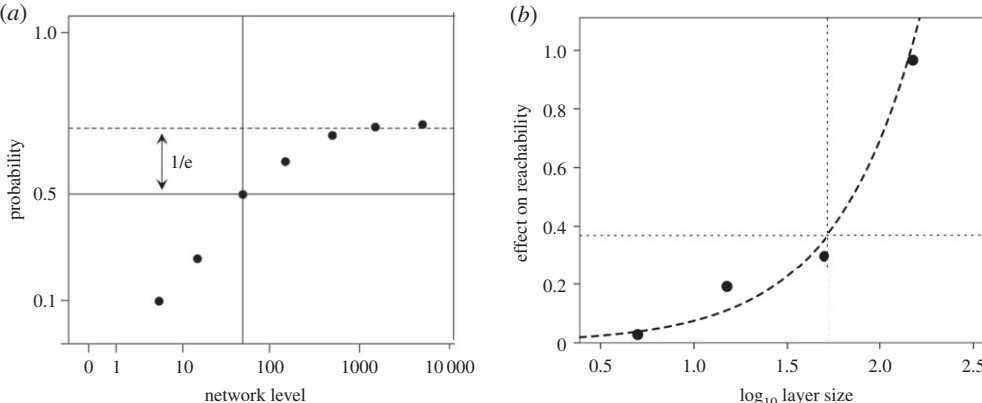

**Figure 5.** (*a*) Cumulative probability of hearing about an item of information through face-to-face contact with an individual who has that knowledge in communities of different size (5, 15, 50, 150, 500, 1500 and 5000), based on the contact rates shown in figure 3 extrapolated out to higher layers. The model assumes that 1% of all community members are initially seeded with the information. The cumulative probability asymptotes at a network size of approximately 150, but the optimum number of alters is 50 (identified by the point at which the curve begins to decelerate, defined by $e^{-1}$ of the way down from the asymptote) since after this, the benefits of increasing network size diminish exponentially. Reproduced from [130]. (*b*) Effect on reachability of removing nodes in different layers of egocentric twitter graphs: the larger the effect, the greater the disruption on information flow. The horizontal dotted lines demarcate 1/*e*th from the asymptote, and the vertical dotted line the optimal group size for information diffusion. Data from [133].

Penetrance was slower if there were fewer cultural 'parents', for example, or if the population size was larger.

To explore the rate at which information flows through a community, Dunbar [130] modelled the likelihood that Ego (at the centre of the network) would hear about a novel innovation via direct face-to-face contact for communities of different size (5, 15, 50, 150, 500, 1500 and 5000 individuals), given the layer-specific rates of contact shown in figure 3 (extrapolated out to the 5000 layer). The probability, $P_i$, of hearing about an innovation seeded somewhere in a network with $i$ layers is the conjoint probability of encountering any given individual in a network layer of size $n_i$ and the likelihood that any one of these individuals would have the trait in question (i.e. be infected), summed across all layers

$$P_i = \Sigma_1{}^i(n_i * r_K) * c_{i[F2F]}, \tag{6.1}$$

where $n_i$ is the number of individuals in the $i$th annulus (network layer), $r_K$ is the likelihood that any one of them will have the trait (here taken to be constant, and equivalent to $r_K = 0.01$), $c_{i[F2F]}$ is the likelihood of contacting any given individual face-to-face. Figure 5*a* plots the results.

The probability of acquiring information reaches an asymptotic value at a community size of approximately 1500, with no further gain in the likelihood of hearing about an innovation as community size increases beyond this. The optimal community size for information transmission can be identified by the inflection point (the point at which the marginal gain begins to diminish). With a graph of this form, this occurs at the value on the X-axis when the asymptotic value on the Y-axis is reduced by 1/e. This is at a community size of exactly 50. The gains to be had by increasing community size beyond approximately 50 diminish exponentially and become trivial beyond a community size of approximately 150 individuals. This was later confirmed by Arnaboldi *et al*. [133] who modelled information diffusion in actual Twitter networks. Figure 5*b* plots the effect on reachability of removing nodes in different layers in a conventional cascade diffusion model. In this case, the inflection point (1/e up from the baseline) is at a layer size of 52.5, very close to the value in figure 5*a* for face-to-face data.

Importantly, and contrary to Granovetter's [134] well-known claim, it seems that it is the inner layers (stronger ties) that may have most impact on the likelihood of acquiring information by diffusion, not the outermost layer (weak ties). The outermost 150-layer (which is disproportionately populated by distant kin [59,135]) presumably serves some other function such as a mutual support network [133]. This finding appears to conflict with earlier analyses [6,137] that have emphasized the importance of weak links (long-range connections) in the rate at which infections or information are propagated in networks. The issue, however, hinges on which layers are counted as strong (short-range) and which weak (long-range). Previous analyses, including Granovetter himself, have tended to be unspecific on this point. If what he had in mind as weak ties was the 50-layer, then his claim holds; if he was thinking of the 150- or even 5000-layer, then it seems he was wrong. Even so, it seems that the information value of 50-layer ties is considerably less than that of alters in the 5- and 15-layers, who are also likely to be considered more trustworthy sources of information. Nonetheless, Granovetter might still be right if either of two conditions hold. One is that the analyses considered only Ego acquiring information by direct personal contact; the models did not consider the impact of upward information flow through the network from the source of the information towards Ego. The other is that Granovetter might have been right for the wrong reason: the function of networks is not information flow (or acquisition) but the provision of direct functional support such as protection against external threats or sources of economic support (a view which would accord better with the view of primate social systems elaborated in §4). In other words, as is the case in primate social systems, information flow is a *consequence* of network structure, not its driving force in terms of evolutionary selection [39].

It may, nonetheless, be that the 150-layer provides the principal access channels to the global network beyond the individual's primary personal social sphere. This is suggested by an analysis that used $M$-polynomials derived from chemical graph theory to integrate Dunbar graphs into the Milgram small world 'six degrees of separation' phenomenon. The capacity to reach an unknown remote individual in 4–6 links is only possible if, at each step in the chain, the message-holder can access a large number of alters in their personal network [67]. However, this analysis only considered 15 versus 150 network contacts, and 150 significantly over-engineers the solution. Further work is needed to explore the optimal network size and structure for transmission in more detail. A moot point, of course, is whether the capacity to send letters to a remote stranger is ever of any real functional value to us, or simply an amusing but unimportant by-product of the way personal networks interface with each other in global networks.

One complicating aspect of real social networks not usually considered in these models is the fact that social subnetworks are characterized by a high level of homophily, especially in the inner layers. In other words, people's friends tend to resemble them on an array of social and cultural dimensions [118,119,123]. Analysing a large (20 million users) cellphone dataset, Miritello *et al.* [85] differentiated, on the basis of calling patterns, two distinct phenotypes: 'social keepers' who focused their calls on a small, stable cohort of contacts (introverts?) and 'social explorers' who had a larger number of contacts with high churn (social butterflies, or extraverts?). Each tended to exhibit a strong preference for its own phenotype. Assuming that phone contact rates mirror face-to-face contact rates (as, in fact, seems to be the case [68,133,138]), explorers were more likely to contact an infected individual because they were more wide-ranging in their social contacts. Keepers remained buffered by their small social radius for longer. This reinforces the suggestion made earlier that innovations frequently go extinct in structured networks because they get trapped in eddies created by network structuring and risk going extinct before they can escape into the wider world.

The role of extraverts in facilitating information flow was also noted by Lu *et al.* [139,140] in a study of networks parametrized by personality-specific contact rates from the community studied by [125]. They found that information flow was more efficient if the network consisted of two distinct phenotypes (in this case, actual introvert and extravert personality types) than if all community members were of the same phenotype. In large part, this was because extraverts (those who opted to prioritize quantity over quality of relationships) acted as bridges between subnetworks, thereby allowing information to flow more easily through the network as a whole.

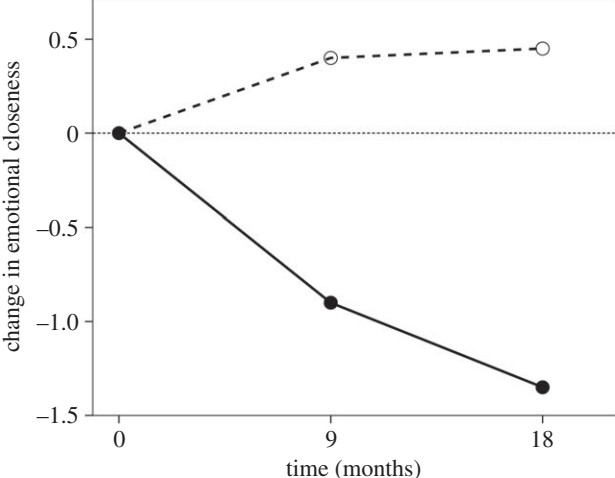

**Figure 6.** Change in the mean emotional closeness (on a 1–10 linear scale) to all members of the extended family (kin) and the original set of friends at time 0 after Ego moves to another city and cannot meet up with them so easily. Family relationships are extremely robust to not being contacted, but friendships decline in emotional closeness very quickly if they are not continuously reinforced. Redrawn from [141].

## 7. Some social consequences

Much of the focus in network dynamics has been on disease propagation. In most models, networks are assumed to remain essentially static in structure over time. This may not always be the case, since network structure may itself respond to both internal threats (stress or deception) and external threats (such as disease or exploitation or attack by outsiders). This is because threats such as disease or exploitation cause a breakdown in trust and trust is, as we saw, central to the structure of social networks. Other factors that might cause networks to restructure themselves include a reduction in the time available for social interaction, access to a sufficiently large population to allow choice [128] or a change in the proportion of phenotypes (sex, personality or family size) when these behave differently. Methods for studying networks that change dynamically through time have been developed [142], although in practice these typically reflect past change rather than how networks are likely to respond to future challenges. Here, my concern is with how networks might change as a consequence of the internal and external forces acting on them.

Because of the way relationships are serviced in social networks (§4), a reduction in time devoted to a tie results in an inexorable decline in the emotional strength of a tie, at least for friendships (figure 6). Note, however, that family ties appear to be quite robust in the face of lack of opportunity to interact. Figure 6 suggests that this effect happens within a matter of a few months (see also [143,144]). Saramäki et al. [145] reported a turnover of approximately 40% in the network membership of young adults over an 18-month period after they had moved away from their home town, most of which occurred in the first nine months. A similar effect will occur when there is a terminal breakdown in a relationship. These seem to occur with a frequency of about 1% of relationships per year, though it is clear that some people are more prone to relationship breakdown than others [146]. Most such breakdowns occur because of a breakdown in trust [59,146].

Although almost never considered in models of network dynamics, the division between family and friends can have significant consequences for the dynamics of networks, especially when comparing natural fertility (no contraception) with demographic transition fertility regimes (those that actively practise contraception). Friendships require significantly more time investment than family relationships to maintain at a constant emotional intensity, especially so in

the outer layers [147,148], and because of this are more likely to fade (and to do so rather quickly) if contact is reduced [147] (figure 6). Family relationships, on the other hand, are more forgiving of breaches of trust and underinvestment. In addition, when family relationships breakdown, they are apt to fracture catastrophically and irreparably [146], creating structural holes. By contrast, most friendships die quietly as a result of reduced contact, in many cases because they are simply replaced by alternatives. This probably means that, when a network is under threat, friendship ties are more likely to be lost than family ties. This would seem to be born out by casual observation of the response to the COVID-19 lockdown: virtual zoom-based family gatherings seem to be much more common than friendship-based meetings.

Under normal circumstances, the gaps left by the loss of a tie following a relationship breakdown are filled by moving someone up from a lower layer or by adding an entirely new person into the network from outside. Saramäki *et al*. [145] noted that, when this happens, the new occupant of the vacated slot is contacted with exactly the same frequency as the previous occupant, irrespective of who they are. It seems that individuals have a distinctive social fingerprint, and this fingerprint is very stable across time [145]. However, if the opportunity for social interaction is restricted, or there is widespread breakdown in the level of trust (as when many people cease to adhere to social rules, or a culture of deception or antisocial behaviour evolves), then the inevitable response is for networks to fragment as individuals increasingly withdraw from casual contacts and focus their attention on those whom they trust most (normally the alters in the innermost layers of their network).

Iñiguez *et al*. [149] and Barrio *et al*. [150] modelled the effect of two kinds of deception (selfish lies versus white lies) on network structure. Selfish lies are those that benefit the liar, while white lies are those that benefit either the addressee or the relationship between the liar and the addressee (e.g. 'Likes' on digital media). These two phenotypes differ radically in the effect they have on the relationship between the individuals concerned: the first will cause a reduction in the frequency of contact resulting in a fragmentation of the network, whereas the second often reinforces network cohesion. If networks shrink sufficiently under stress, they may invert (figure 4).

There is indirect evidence for this in the effect that parasite load has on the size of communities in tribal societies: these decline in size the closer they are to the equator (the tropics being the main hotspot for the evolution of new diseases), and this correlates in turn with a corresponding increase in the number of languages and religions, both of which restrict community size [151,152]. At high latitudes, where parasite loads tend to be low and less stable climates make long-range cooperation an advantage, community sizes are large, people speak fewer languages and religions tend to have wider geographical coverage, which, between them, will result in more extensive global networks [151,152].

Similar effects have been noted in financial networks, where network structure between institutions that trade with each other also depends on trust. There has been considerable interest in how network structure might influence the consequences of contagion effects when financial institutions collapse. Network structure can affect how shocks spread through networks of banks, giving rise to default cascades in ways not dissimilar to the way diseases propagate through human social networks. Although well-connected banks may be buffered against shocks because of the way the effects are diluted [153–155] much as a well-connected individuals may be buffered against social stresses, a loss of trust between institutions invariably results in the contraction of networks, associated with more conservative trading decisions and a greater reluctance to lend [156] in ways reminiscent of social networks fragmenting in the face of a loss of trust.

If effective network size (i.e. the number of ties an individual has) is reduced as a result of such effects, more serious consequences may follow at the level of the individual for health, wellbeing and even longevity. Smaller social networks are correlated with increasing social isolation and loneliness, and loneliness in turn has a dramatic effect on morbidity and mortality rates. There is now considerable evidence that the number and quality of close friendships that an individual has directly affects their health, wellbeing, happiness, capacity to recover from surgery and major illness, and their even longevity (reviewed in [96,157]), as well as their engagement with, and trust

in, the wider community within which they live [114,115]. Indeed, the effects of social interaction can even outweigh more conventional medical concerns (obesity, diet, exercise, medication, alcohol consumption, local air quality, etc.) as a predictor of mortality [158]. Most epidemiological studies have focused on close friends, but there is evidence that the size of the extended family can have an important beneficial effect, especially on children's morbidity and mortality risks [159]. These findings are mirrored by evidence from primates: the size of an individual's intimate social circle has a direct impact on its fertility, survival, how quickly it recovers from injury, and ultimately its biological fitness [160–166].

It is worth noting that Dunbar graphs, with their basis in trust, have been used to develop online 'secret handshake' security algorithms for use in pervasive technology (e.g. *Safebook* [167,168]). Pervasive technology aims to replace cellphone masts by using the phones themselves as waystations for transmitting a call between sender and addressee. The principal problem this gives rise to is trust: a phone needs to be able to trust that the incoming phone is not intent on accessing its information for malicious purposes. *Safebook* stores the phone owner's Seven Pillars as a vector which can then be compared with the equivalent vector from the incoming phone. A criterion can be set as to how many 'pillars' must match for another phone to be considered trustworthy. The Dunbar graph has also been used to develop a bot-detection algorithm by comparing a node's network size and shape with that of a real human (i.e. a Dunbar graph): this algorithm out-performs all other currently available bot-detection algorithms [169].

We might ask what effects we might expect in the light of this from the lockdowns imposed by most countries in 2020 in response to COVID-19. I anticipate four likely effects. One is that if lockdown continues for more than about three months, we may expect to see a weakening of existing friendships, especially in groups like the elderly whose network sizes are already in age-dependent decline. Since older people find it more difficult to make new friends, an increased level of social isolation and loneliness is likely to result, with consequent increases in the diseases of old age, including dementia and Alzheimer. Second, we may expect to see an increased effort to recontact old friends, in particular, immediately after lockdown is lifted. We already see evidence for this in telephone call patterns: if there is a longer than normal gap before an alter is called again, the next call is significantly longer than average as though attempting to repair the damage to relationship quality [170]. Third, the weakening of friendship quality can be expected (i) to make subsequent meetings a little awkward because both parties will be a little unsure of how they now stand when meeting up again and (ii) to result in some churn in networks where new friendships developed through street-based COVID community groups are seen as more valuable (and more accessible) than some previous lower rank friendships. Finally, we may expect the fear of coronavirus contagion (an external threat) to result in a reduction in the frequency with which some individuals (notably introverts and the psychologically more cautious) visit locations where they would come into casual contact with people they do not know. They may also reduce frequencies of contact with low-rank friends, and perhaps even distant family members, whose behaviour (and hence infection risk) they cannot monitor easily. This is likely to result in more inverted networks, as well as networks focused mainly on people who are more accessible. Although this effect will weaken gradually with time, and network patterns are likely to return to pre-COVID patterns within 6–12 months, some friendship ties may have been sufficiently weakened to slip over the threshold into the 500 (acquaintances) layer.

## 8. Conclusion

My aim in this paper has been to introduce a rather different perspective on the social structure of communities than that normally considered in disease propagation models, and to explain the forces that underpin real-world social networks. I have presented evidence to suggest that human social networks are very much smaller and more highly structured than we usually assume. In addition, network size and composition can vary considerably across individuals as a function of sex, age, personality, local reproductive patterns and the size of the accessible population. While casual contacts might be important for the spread of highly infectious diseases [8,10], most

social time is actually devoted to no more than 15 individuals and this will slow down rates of transmission if physical contact or repeated exposure to the same individual is required for successful infection. Both external and internal threats can destabilize network ties by affecting the level of trust, causing networks to contract and fragment. If networks fragment under these threats, there will be knock-on consequences in terms of health, wellbeing and longevity.

Data accessibility.  This article has no additional data.

Competing interests.  I declare I have no competing interests.

Funding.  Most of the research reported here was funded by the UK EPSRC/ESRC TESS research grant, British Academy Centenary Research Project (Lucy to Language), the ERC RELNET Advanced research fellowship, the EU FP7 Socialnets grant and the EU Horizon 2020 IBSEN research grant.

Acknowledgements.  I thank the reviewers for drawing my attention to some useful additional material.

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
