## [Reviewer comments · Proceedings. Mathematical, Physical, and Engineering Sciences]

Review History

RSPA-2020-0446.R0 (Original submission)

Review form: Referee 1

Is the manuscript an original and important contribution to its field?

Acceptable

Is the paper of sufficient general interest?

Good

Is the overall quality of the paper suitable?

Good

Can the paper be shortened without overall detriment to the main message?

Yes

Do you think some of the material would be more appropriate as an electronic appendix?

No

Do you have any ethical concerns with this paper?

No

Recommendation?

Accept with minor revision (please list in comments)

Comments to the Author(s)

This manuscript provides an interesting review and commentary of the psychological and behavioural determinants of social network structure in humans and primates. The manuscript is of the form of a review/synthesis of previous research, rather than presenting any novel findings.

Section 2 overviews the common structure of human and primate networks, and the behavioural and psychological determinants of this structure.

Section 3 overviews that human social networks are layered, and that this layered structure is consistent across different media, and that this also applies to primate networks. It also considers some of the determinants of heterogeneity in human social networks – including sex, age, and the distinction between family and friend ties.

Section 4 explains the bonding process in primates. It documents the role of grooming, and the ways that humans have managed to expand their number of contacts they ‘groom’ by undertaking other activities that trigger the same endorphin system. It also explains the cognitive mechanism underlying bonding in primates – being able to predict and rely upon another’s behaviour, and how this leads to homophily in social networks.

Section 5 overviews the way in which time constraints determine the structure of social networks. It surveys different models that aim to recreate the structure of human social networks.

Section 6 surveys how this structure affects the diffusion of e.g. information.

Section 7 looks at some of the social implications of this social network structure.

Specific comments:

The ‘fractal structure of networks’ needs to be defined more clearly/rigorously.

"They also treat diseases and innovations as bipolar phase states (you either have them or you don't)." - True, but this isn't an issue addressed anyway this manuscript, it would be good to hear the author's thoughts on this.

"Although network analysts began to appreciate quite early on that human populations are highly structured and that this structure could dramatically affect how innovations propagate through networks (6,7) Watts & Stogatz 1998; Keeling 1999), only very recently has it been appreciated that the structure created by social or spatial organisation might affect the speed with which diseases or information propagate through a population (8-10) Read et al. 2008; Danon et al. 2013; Read et al. 2014)." - I don't think I understand the difference between these two situations & both of the early papers focus on disease spread

(On the distinction between family and friends). "Although almost never considered in models of network dynamics, this division can have significant consequences for the dynamics of networks, especially between countries characterised by small and large families." Household models are epidemiological models built with this consideration in mind – the distinction between within-household interactions and between-household interactions. There is a wealth of literature on such models (see e.g. House and Keeling (2009), Pellis Ferguson and Fraser (2009)). In addition, many of the large simulation models combine households with social networks.

"A comparison of migrant versus host communities in Spain revealed that there is indeed a phase transition between conventionally structured networks (layers with a concave structure) to

reversed networks (convex structures) at $\mu=0$ (110) Tamarit et al. (104)." The concepts of convex and concave structure need to be defined and explained in more detail for the benefit of the reader.

"Much of the focus in network dynamics has been on disease propagation. In all these studies, networks are assumed to remain essentially static in structure over time." Many epidemic network models also consider dynamic network structures, (see e.g. Masuda and Holme (2017) for an overview of some recent methods).

Review form: Referee 2

Is the manuscript an original and important contribution to its field?

Good

Is the paper of sufficient general interest?

Excellent

Is the overall quality of the paper suitable?

Good

Can the paper be shortened without overall detriment to the main message?

Yes

Do you think some of the material would be more appropriate as an electronic appendix?

No

Do you have any ethical concerns with this paper?

No

Recommendation?

Accept with minor revision (please list in comments)

Comments to the Author(s)

See attached file

Decision letter (RSPA-2020-0446.R0)

12-Jun-2020

Dear Professor Dunbar

The Reviews Editor of Proceedings A has now received comments from referees on the above paper and would like you to revise it in accordance with their suggestions which can be found below (not including confidential reports to the Reviews Editor). Please submit a copy of your revised paper within four weeks.

To revise your manuscript, log into <http://mc.manuscriptcentral.com/prsa> and enter your Author Centre, where you will find your manuscript title listed under "Manuscripts with Decisions." Under "Actions," click on "Create a Revision." Your manuscript number has been appended to denote a revision.

You will be unable to make your revisions on the originally submitted version of the manuscript. Instead, revise your manuscript and upload a new version through your Author Centre.

When submitting your revised manuscript, you will be able to respond to the comments made by the referee(s) and upload a file "Response to Referees" in "Section 6 - File Upload". Please use this to document how you have responded to the comments, and the adjustments you have made. In order to expedite the processing of the revised manuscript, please be as specific as possible in your response to the referee(s).

IMPORTANT: Your original files are available to you when you upload your revised manuscript. Please delete any unnecessary previous files before uploading your revised version.

Once again, thank you for submitting your manuscript to Proc. R. Soc. A and I look forward to receiving your revision. If you have any questions at all, please do not hesitate to get in touch.

Yours sincerely
 Raminder Shergill
 proceedingsa@royalsociety.org

Reviewer(s)' Comments to Author:
 Referee: 1

Comments to the Author(s)

This manuscript provides an interesting review and commentary of the psychological and behavioural determinants of social network structure in humans and primates. The manuscript is of the form of a review/synthesis of previous research, rather than presenting any novel findings.

Section 2 overviews the common structure of human and primate networks, and the behavioural and psychological determinants of this structure.

Section 3 overviews that human social networks are layered, and that this layered structure is consistent across different media, and that this also applies to primate networks. It also considers some of the determinants of heterogeneity in human social networks – including sex, age, and the distinction between family and friend ties.

Section 4 explains the bonding process in primates. It documents the role of grooming, and the ways that humans have managed to expand their number of contacts they 'groom' by undertaking other activities that trigger the same endorphin system. It also explains the cognitive mechanism underlying bonding in primates – being able to predict and rely upon another's behaviour, and how this leads to homophily in social networks.

Section 5 overviews the way in which time constraints determine the structure of social networks. It surveys different models that aim to recreate the structure of human social networks.

Section 6 surveys how this structure affects the diffusion of e.g. information.

Section 7 looks at some of the social implications of this social network structure.

Specific comments:

The 'fractal structure of networks' needs to be defined more clearly/rigorously.

"They also treat diseases and innovations as bipolar phase states (you either have them or you don't)." - True, but this isn't an issue addressed anyway this manuscript, it would be good to hear the author's thoughts on this.

"Although network analysts began to appreciate quite early on that human populations are highly structured and that this structure could dramatically affect how innovations propagate through networks (6,7) Watts & Stogatz 1998; Keeling 1999), only very recently has it been appreciated that the structure created by social or spatial organisation might affect the speed with which diseases or information propagate through a population (8-10) Read et al. 2008; Danon et al. 2013; Read et al. 2014)." - I don't think I understand the difference between these two situations & both of the early papers focus on disease spread

(On the distinction between family and friends). "Although almost never considered in models of network dynamics, this division can have significant consequences for the dynamics of networks, especially between countries characterised by small and large families." Household models are epidemiological models built with this consideration in mind – the distinction between within-household interactions and between-household interactions. There is a wealth of literature on such models (see e.g. House and Keeling (2009), Pellis Ferguson and Fraser (2009)). In addition, many of the large simulation models combine households with social networks.

"A comparison of migrant versus host communities in Spain revealed that there is indeed a phase transition between conventionally structured networks (layers with a concave structure) to reversed networks (convex structures) at $\mu=0$ (110) Tamarit et al. (104)." The concepts of convex and concave structure need to be defined and explained in more detail for the benefit of the reader.

"Much of the focus in network dynamics has been on disease propagation. In all these studies, networks are assumed to remain essentially static in structure over time." Many epidemic network models also consider dynamic network structures, (see e.g. Masuda and Holme (2017) for an overview of some recent methods).

Referee: 2

Comments to the Author(s)
See attached file

Author's Response to Decision Letter for (RSPA-2020-0446.R0)

See Appendix A.

Decision letter (RSPA-2020-0446.R1)

10-Jul-2020

Dear Professor Dunbar

On behalf of the Editor, I am pleased to inform you that your manuscript entitled "Structure and Function in Human and Primate Social Networks: Implications for Diffusion, Network Stability and Health" has been accepted in its final form for publication in Proceedings A.

Our Production Office will be in contact with you in due course. You can expect to receive a proof of your article soon. Please contact the office to let us know if you are likely to be away from e-mail in the near future. If you do not notify us and comments are not received within 5 days of sending the proof, we may publish the paper as it stands.

Open access

You are invited to opt for open access, our author pays publishing model. Payment of open access fees will enable your article to be made freely available via the Royal Society website as soon as it is ready for publication. For more information about open access please visit http://royalsocietypublishing.org/site/authors/open_access.xhtml. The open access fee for this journal is £1700/\$2380/€2040 per article. VAT will be charged where applicable.

Note that if you have opted for open access then payment will be required before the article is published – payment instructions will follow shortly. If you wish to opt for open access then please inform the editorial office (proceedingsa@royalsociety.org) as soon as possible.

Under the terms of our licence to publish you may post the author generated postprint (ie. your accepted version not the final typeset version) of your manuscript at any time and this can be made freely available. Postprints can be deposited on a personal or institutional website, or a recognised server/repository. Please note however, that the reporting of postprints is subject to a media embargo, and that the status the manuscript should be made clear. Upon publication of the definitive version on the publisher's site, full details and a link should be added.

You can cite the article in advance of publication using its DOI. The DOI will take the form: 10.1098/rspa.XXXX.YYYY, where XXXX and YYYY are the last 8 digits of your manuscript number (eg. if your manuscript number is RSPA-2017-1234 the DOI would be 10.1098/rspa.2017.1234).

For tips on promoting your accepted paper see our blog post:
<https://blogs.royalsociety.org/publishing/promoting-your-latest-paper-and-tracking-your-results/>

Thank you for your submission. On behalf of the Editors of the journal, we look forward to your continued contributions to the Journal.

Best wishes
Raminder Shergill,
Proceedings A Editorial Office
proceedingsa@royalsociety.org

Appendix A

REVIEWER 1

This paper argues that human social networks are both more structured and smaller than is apparent from casual observation. The paper also notes that structure and size matter for disease and information transmission. An excellent survey of the literature on the size of social networks and the consistency of conclusions about network structure and size is provided. For the type of networks considered in this paper the points about structure and size are well supported. The argument that human social networks are not the simple networks analyzed in most of the simple models of disease transmission (such as branching processes) is compelling. It might be valuable to point out that those simple models are the basis for the critical R_0 value that is so commonly used in public discussions of epidemics.

Comment added to Introduction.

But it's also useful to note that models of disease transmission such as SIR, SIS and SIRS models can be applied to any network. The discussion in this paper is primarily focused on friendship networks (using a broad definition of friendship). For contagion of ideas, customs, fashion and the like this seems an appropriate definition of the relevant network upon which to study contagion. It's less clear that it leads to the appropriate definition of the network in which to study disease contagion. This surely depends on the disease, but for one which can apparently spread from person to person based on proximity, it doesn't seem appropriate. Casual contacts in a store or on a subway can spread the disease while friendships which don't currently involve physical interaction can't spread the disease.

This was actually pointed out, but I have emphasised it.

Minor comments:

- There is a literature at the intersection of economics/finance and computer science that is relevant to endogeneity of networks and what is called in that literature "systemic risk"—usually meaning a cascade of failures. See Benoit et al (Review of Finance 2017) and Glasserman and Young (Journal of Economic Literature 2016) for recent surveys. This is relevant to both the endogeneity of networks and contagion on those networks.

Interesting point. Paragraph added on bank networks and failure cascades.

- It would be useful to comment on the relation between the structure of networks considered here and the observations about the importance of 1 long range ties as in Strogatz and Watts (Nature 1998) and Kleinberg (Nature 2000).

I have added a comment on this very relevant point.

REVIEWER 2

This manuscript provides an interesting review and commentary of the psychological and behavioural determinants of social network structure in humans and primates. The manuscript is of the form of a review/synthesis of previous research, rather than presenting any novel findings.

Section 2 overviews the common structure of human and primate networks, and the behavioural and psychological determinants of this structure.

Section 3 overviews that human social networks are layered, and that this layered structure is consistent across different media, and that this also applies to primate networks. It also considers some of the determinants of heterogeneity in human social networks – including sex, age, and the distinction between family and friend ties.

Section 4 explains the bonding process in primates. It documents the role of grooming, and the ways that humans have managed to expand their number of contacts they ‘groom’ by undertaking other activities that trigger the same endorphin system. It also explains the cognitive mechanism underlying bonding in primates – being able to predict and rely upon another’s behaviour, and how this leads to homophily in social networks.

Section 5 overviews the way in which time constraints determine the structure of social networks. It surveys different models that aim to recreate the structure of human social networks.

Section 6 surveys how this structure affects the diffusion of e.g. information.

Section 7 looks at some of the social implications of this social network structure.

Specific comments:

The ‘fractal structure of networks’ needs to be defined more clearly/rigorously.

I have reworded the relevant sentences to clarify.

"They also treat diseases and innovations as bipolar phase states (you either have them or you don't)." - True, but this isn't an issue addressed anyway this manuscript, it would be good to hear the author's thoughts on this.

This has now been deleted.

"Although network analysts began to appreciate quite early on that human populations are highly structured and that this structure could dramatically affect how innovations propagate through networks (6,7) Watts & Stogatz 1998; Keeling 1999), only very recently has it been appreciated that the structure created by social or spatial organisation might affect the speed with which diseases or information propagate through a population (8-10) Read et al. 2008; Danon et al. 2013; Read et al. 2014)." - I don't think I understand the difference between these two situations & both of the early papers focus on disease spread

Reworded to clarify.

(On the distinction between family and friends). "Although almost never considered in models of network dynamics, this division can have significant consequences for the dynamics of networks, especially between countries characterised by small and large families." Household models are epidemiological models built with this consideration in mind – the distinction between within-household interactions and between-household interactions. There is a wealth of literature on such models (see e.g. House and Keeling (2009), Pellis Ferguson and Fraser (2009)). In addition, many of the large simulation models combine households with social networks.

I have reworded this section to clarify the substantive issues and added these two useful references.

"A comparison of migrant versus host communities in Spain revealed that there is indeed a phase transition between conventionally structured networks (layers with a concave structure) to reversed networks (convex structures) at $\mu=0$ (110) Tamarit et al. (104)." The concepts of convex and concave structure need to be defined and explained in more detail for the benefit of the reader.

Reworded to clarify

"Much of the focus in network dynamics has been on disease propagation. In all these studies, networks are assumed to remain essentially static in structure over time." Many epidemic network models also consider dynamic network structures, (see e.g. Masuda and Holme (2017) for an overview of some recent methods).

Reworded and reference added.

I have removed Figure 7.

=====

REVIEWER 3

I'm afraid I don't have time to do a proper review. Oh well... my apologies.

Nevertheless, I want to help and so, in spite of myself, I skimmed Prof. Dunbar's paper at breakfast. I'd heard of his famous Dunbar number (150), and was pleased to find he's a very graceful writer. His paper is well organized, intellectually wide ranging, and packed with interesting information (e.g., I learned that stroking my dog at the right frequency is thought to activate the release of endorphins in both him and me, via a known neural pathway). Anyway, I have nothing substantive to offer here, except that I think the paper is interesting and tenable as is. Obviously, referees could quibble about this or that, but it seems fine to me.